# Changes in Family Spirituality in Response to Family Intervention Utilizing the Family Life Review

**DOI:** 10.3390/nursrep15080277

**Published:** 2025-07-30

**Authors:** Naohiro Hohashi, Haruka Yano

**Affiliations:** Division of Family Health Care Nursing, Department of Nursing, Graduate School of Health Sciences, Kobe University, Kobe 654-0142, Japan; haruka-yano@kashibaseiki.fujiikai.jp

**Keywords:** family nursing, family spirituality, family life review, concentric sphere family environment theory

## Abstract

**Background/Objectives**: Family spirituality is an important concept in family nursing that reinforces the meaning of the family’s existence. However, no studies on specific family intervention methods have been conducted to date. The purpose of this study was to verify the effect of family interventions using the family life review (FLR) program on changes to family spirituality. **Methods**: An FLR was conducted on six families having older adult members and undergoing family spiritual suffering, with two sessions spaced one week apart. The FLR was conducted using the Plot of Family Story (PFS), a tool for reviewing family history based on the concentric sphere family environment theory (CSFET). Semi-structured interviews and scoring using the Family Spirituality Index were conducted based on CSFET at three points in time: first before, and then after, the FLR, then again one month later, and changes in family spirituality were analyzed using mixed methods. **Results**: The families encountered family spiritual suffering in the family internal environment system, family system unit, and chrono system according to the CSFET. The FLR, when used with the PFS, was able to maintain, enhance, confer, and actualize family spirituality corresponding to the spiritual suffering being experienced by the target families. **Conclusions**: Family intervention through an FLR using the PFS can improve family spirituality in families undergoing family spiritual suffering. In this study, the PFS became a legacy for the family and raised awareness of the value of their existence.

## 1. Introduction

Spirituality has been attracting more attention as an element related to human health and quality of life [1], but previous studies have mostly focused on individual spirituality. Individual spirituality incorporates various attributes but is expressed as the meaning and purpose of life and the significance of one’s existence, with life review being one way of supporting individual spirituality. Life review is a necessary process to create an opportunity to connect the present, past, and future and achieve an integration of the ego. Its effects include dealing with current and past problems; discovering the meaning and continuity of life; and reconciliation with family and friends [2,3]; it is one of the most popular methods of providing support for people suffering spiritual pain. The life review provides an opportunity to relive and re-evaluate past experiences, and, by reviewing one’s past social roles, achievements, and contributions to family and society, one can recognize the value of one’s existence, accept the current situation, and sense the meaning of life [4]. Previous studies have shown that the life review is effective in improving an individual’s spiritual well-being, improving self-esteem, and reducing depression [4].

Spirituality is believed to exist not only in individuals but also in families, but little research has been conducted on family spirituality. For example, although some studies have shown that family spirituality strengthens family health [5] and that the role of spirituality in families is clearly an important factor in family resilience [6], the definition of family spirituality has not been consistent. Based on the concentric sphere family environment theory (CSFET) [7], a middle-range family nursing theory, Hohashi defines family spirituality as “a set of core beliefs of the family system unit that recognizes the meaning of the existence of the family system unit in a transcendental dimension” [8]. He also classifies family spirituality into the six systems of the CSFET (the family internal environment system, family system unit, chrono system, micro system, macro system, and supra system). Core family beliefs are beliefs held among family members that are commonly recognized by all family members and that possess a high degree of universality. Core beliefs are causal family beliefs that exist in the family’s subconscious and that give rise to and change multiple intermediary family beliefs. Hohashi also describes family spiritual suffering as family symptoms/signs [8], i.e., “the suffering experienced when the spirituality of the family system unit is threatened.” Family spirituality improves family well-being and functioning, strengthens the ability to cope with crises, and reinforces the family’s purpose and meaning, so support for family spirituality is an important aspect of family nursing [9].

Life reviews can also be applied to families, and conducting life reviews on families is said to clarify family problems and promote reconciliation within the family [2]. O’Hora developed a systematic Family Life Review (FLR) program as a form of family therapy [10]. Empirical research has revealed that the FLR, in which families reflect on their history of family experiences and events, impacts relationships by strengthening family ties and promoting intergenerational bonds. In addition, families have described the importance of sharing the depth of meaning of being a family, the mutuality of relationships, self-worth, and legacy (something that allows the family to relive past events), and have become strongly aware of the value of family by sharing memories between and across generations, looking back on the past, and remembering one another, creating something that will be passed down within the family [11]. Family spirituality is a core family belief that recognizes the purpose and meaning of the family; therefore, FLR can be applied to support spiritual suffering in families. However, no studies have been reported using FLR for the purpose of supporting family spirituality.

Life reviews require the re-evaluation of past experiences. One tool for looking back on the past is the Plot of Family Story (PFS), which was developed by Hohashi, based on the CSFET [12]. The PFS is a chronological table indicating the relationship and process of the plot of family, family drama, family story, the degree of family events, and the condition of changes in the family [12]. The PFS is also a tool for looking back on family history, and by tabulating when events occurred and what events the family experienced, the family’s perception of the events, and the changes in the family due to such events, this affords the ability to visualize the extent of and the ways in which the family plot and family drama have influenced family symptoms/signs. We believe that using PFS in family FLR programs can help families to reevaluate the past and reaffirm the value of the family, and, therefore, that PFS can be used in FLR programs to support families with family spiritual suffering. Hohashi [8] describes family spirituality support as “family belief support for family beliefs that recognize the meaning of the family’s existence in order to enhance the meaning of the family’s existence.” Consequently, we believe that the FLR process of looking back on one’s experiences of dealing with various family events and reconsidering the value of the family’s existence is able to serve as family spirituality support.

Based on the above, this study targeted families who were undergoing family spiritual suffering and aimed to clarify the ways in which family spirituality changes before and after family intervention through FLR methods using PFS, and to verify its usefulness in family spirituality support.

### Operational Definitions of Terms

Family spirituality: A set of core beliefs of the family system unit that recognizes the meaning of the existence of the family system unit within a transcendental dimension [8]Family spirituality support: Family belief support for family beliefs that recognize the meaning of the family’s existence, in order to enhance the meaning of the family’s existence (actualization of family spirituality, conversion of family spirituality, enhancement of family spirituality, conferment of family spirituality, uniformity of family spirituality, and maintenance of family spirituality) [8]Actualization of family spirituality: An intervention that transforms the subconscious family spirituality of all family members into an overt family spirituality [8]Conversion of family spirituality: An intervention that transforms negative family spirituality into positive family spirituality [8]Enhancement of family spirituality: An intervention that further elevates the degree of positive family spirituality [8]Conferment of family spirituality: An intervention that enables the family to possess a new, positive family spirituality that it had not previously possessed [8]Uniformity of family spirituality: An intervention that changes family members’ spirituality perceived by some family members into family spirituality that is perceived by all family members [8]Maintenance of family spirituality: An intervention that continues to maintain positive family spirituality [8]Family internal environment system: The family environment system that exists within the family system unit, which is the area within the family system unit where individual family members interact with each other [7]Family system unit: Another term for family, used to clarify that the family is both a system and a unit [7]Chrono system: A concept to indicate the process of temporal change and the transformation of the family internal environment system, family external environment system, and family system unit in a time frame moving from the past to the future [7]Micro system: A familiar area in the neighborhood of the family system unit, based on comprehensive physical/objective and psychological/subjective assessments [7]Macro system: The family members’ sphere of daily activities that is distant from the family system unit, based on comprehensive physical/objective and psychological/subjective assessments [7]Supra system: The outer frame that creates the family environment system, which is directly or indirectly related to other family environment systems, and encompasses the family environment in its entirety [7]

## 2. Methods

### 2.1. Research Design

This study was conducted as a quasi-experimental study, with a before-and-after comparative design in the intervention group.

### 2.2. Target Families

Referring to research in previous studies [11,13], this study targeted families with older adult family members who had experienced the death of a family member, a family member with a difficult-to-treat illness, or separation from a family member. The target families were recruited in a city in Japan through two visiting nurse stations that cooperated with the study; from among the families expressing a willingness to participate in the study, those who the researchers judged to be experiencing family spiritual suffering were recruited using purposive sampling.

### 2.3. Procedure for Implementing FLR

The FLR consisted of two sessions of approximately 60 min each, based on the structure of the bereavement life review developed by Ando et al. [13]. The interval between the first and second sessions was approximately one week, similar to previous studies on bereavement life reviews and life reviews [4,13].

Table 1 shows examples of the questions used in the FLR. In the first session, the family was asked to talk freely about the changes in their family over time using the PFS. In the second session, the family was asked to review the PFS they had completed in the first session and to integrate the FLR. The contents of the FLR were recorded as memos, and the contents were recorded and transcribed verbatim from the audio data.

### 2.4. Method for Evaluating the Effect of the FLR

The effect of the FLR was evaluated through a mixed method using a semi-structured interview on family spirituality and scores from the Family Spirituality Index [14]. The FLR was evaluated at three points: baseline (just before the first FLR), immediately after the second FLR, and approximately one month later. The timing of the evaluation was based on previous research on life reviews, as many previous studies have found that meaningful changes in personal spirituality occur immediately after the FLR [4]. In addition, a study that confirmed the correlation between the passage of time and individual spirituality showed that the effect of life reviews on spirituality continued for up to three months after the life review, but little change was observed between measurements taken one month and three months after the life review [15]. Therefore, in this study, we decided to conduct evaluations approximately one month after the FLR.

#### 2.4.1. Semi-Structured Interview on Family Spirituality

A semi-structured interview on family spirituality was conducted for approximately 120 min. An example of the interview guide is shown below. The content was audio- and video-recorded, and a verbatim transcript was produced from the audio data.

Why do you think your family exists for your family members/the whole family/people, things, and events?What goals and objectives do you think your family has for your family members/the whole family/people, things, and events?Why do you think that is the reason for the existence/purpose and goal of the family?From what I have just said, which one is your family’s understanding of family spirituality (while showing the member the Family Spirituality Index)? Also, why do you think so?

#### 2.4.2. Family Spirituality Index

To evaluate changes in the strength of family spirituality, we used the Family Spirituality Index (Table 2), which is composed of items from a previous study that clarified the attributes of family spirituality [14]. This consists of 21 items and is classified into six systems, based on the CSFET (family internal environment system, family system unit, chrono system, micro system, macro system, and supra system).

Respondents were asked to score the strength of their feelings about each item on a scale ranging from 0 to 10, from “I don’t agree at all (0)” to “I strongly agree (10).” In addition, if their feeling toward an item exceeded 10, it was rated 10+. The Family Spirituality Index was distributed to family members after the semi-structured interview and collected after they had completed it. All participating family members consulted and agreed upon the questionnaire before completing the Family Spirituality Index.

### 2.5. Analysis Method

The FLR was evaluated based on the classification of family spirituality support, using a mixed method of semi-structured interviews on family spirituality and scores from the Family Spirituality Index. When interpretation of the verbatim transcripts, including interactions between family members, was necessary, objective information observed from recorded footage and objective information obtained through researcher observations were added to the analysis.

To ensure the trustworthiness of the analysis and act as an investigator triangulation process [16], the semi-structured interviews were conducted by two researchers who were familiar with family interviews to obtain a comprehensive understanding of each family. To reduce researcher bias, all analyses were performed independently by two family nursing researchers, and any differences of opinion were resolved through discussion in multiple general meetings, with the consideration process carefully repeated until a consensus was reached. In addition, peer debriefings were regularly held with the participation of nine researchers with expertise in family nursing, and the interpretations of the study results were refined through discussion.

### 2.6. Ethical Considerations

This study was conducted after being reviewed and approved by the institutional review board of the affiliated university. The participants were informed of the purpose and methods of the study, the benefits and risks of the study, confidentiality obligations, their freedom to withdraw midway through the study, that there would be no disadvantages for withdrawing midway or not participating, and that they were giving their consent to cooperate with the study by signing a consent form. The information on the target families was anonymized by ID number, using a correspondence table with names to ensure that families and individuals could not be identified. The anonymized data and the correspondence table were stored securely in separate lockable storage cabinets to prevent information leaks.

## 3. Results

### 3.1. Overview of the Target Families

Table 3 indicates an overview of the six target families and their family spiritual suffering. The six families encountered family spiritual suffering in one or more of the following: family internal environment system, family system unit, and chrono system.

### 3.2. Changes in Family Spirituality for Each Family

Changes in family spirituality for each family in the system in which they encountered family spiritual suffering are indicated below, with the narratives of family members being shown in italics.

#### 3.2.1. Changes in Family Spirituality in Family ID 1

Changes in family spirituality are indicated in Table 4. Belief in the family internal environment system items: “1. The family believes that the family exists for the sake of children and grandchildren,” and “2. The family believes that the wishes of family members should be respected” were enhanced immediately after the FLR. Belief in “5. The family believes that it is important for family members to trust spirituality” was enhanced one month after the FLR. The eldest daughter said, “*I tend to regret things, but ultimately I want to reach a state of true gratitude*.” “*Whenever we fight, I look back at this (the PFS)*,” emphasizing that when she feels irritated in her daily life, she wants to look back at the PFS and reaffirm her gratitude to her parents.

Belief in “6. The family believes that family members should support each other” in the family system unit was maintained immediately after the FLR. The eldest daughter said, “*Family members need to support each other. Others won’t support us to that extent*,” and the wife agreed. Belief in “7. The family believes that the family is a kind of microcosm of society” was conferred immediately after the FLR. The eldest daughter said, “*If one family is harmonious and there are many more like it, it will connect to society. I realized that when all family members trust and support each other, they spread their vibrations to the community and society*.” Belief in “8. The family believes that the family is irreplaceable” was maintained immediately after the FLR and enhanced one month after the FLR. The eldest daughter said, “*Family was my source of life*,” and reaffirmed her sense of security and gratitude for the existence of family members. Belief in “11. The family believes that the family is a safe place” was enhanced immediately after the FLR, and the eldest daughter said, “*Family is a place where you can be yourself*.”

#### 3.2.2. Changes in Family Spirituality in Family ID 2

Changes in family spirituality are indicated in Table 5. Belief in the family system unit’s “8. The family believes that the family is irreplaceable” was maintained immediately after the FLR. The eldest daughter said, “*When my mother doesn’t come home from the facility, the house itself is completely lifeless. When my mother comes home, the house is really lively and we can all eat dinner together with my mother*,” and she recognized that the presence of family is irreplaceable and important.

#### 3.2.3. Changes in Family Spirituality in Family ID 3

Changes in family spirituality are indicated in Table 6. Belief in the family system unit’s “6. The family believes that family members should support each other” was maintained immediately after the FLR. The mother and eldest daughter said, “*We believe that family members are able to help each other when they are in trouble*,” reaffirming the family spirituality that had been wavering. In addition, belief in “9. The family believes that the family has a trusting relationship that allows family members to talk to each other” was enhanced one month after the FLR. The eldest daughter said, “*My relationship with my daughter has improved, and she has started to talk to me a little, even if she is somewhat reserved*.”

Belief in the chrono system’s “15. The family believes that they should bequeath their own culture” was actualized immediately after the FLR. Recalling that the family had cherished the fields through the FLR, the eldest daughter said, “*I think that houses, buildings, fields, and the like should not be forgotten*.”

#### 3.2.4. Changes in Family Spirituality in Family ID 4

Changes in family spirituality are indicated in Table 7. Belief in the family system unit items “6. The family believes that family members should support each other,” “8. The family believes that the family is irreplaceable,” “9. The family believes that the family has a trusting relationship that allows family members to talk to each other,” and “11. The family believes that the family is a safe place” were enhanced immediately after the FLR. The wife said, “*I can’t ask others for anything, but my children do it properly. So I’m grateful for that*,” and “*I’m really glad that I’m aware that my family is very fulfilling*.” The husband said, “*It’s the best thing for a family to have someone come rushing over if something happens. I’d be so worried if no one came*,” and “*We respect each other. Now we understand each other*.” The significance of the family, which had been taken for granted until now, was confirmed, and family ties were strengthened by boosting the sense of gratitude toward other family members.

Belief in the chrono system for “12. The family believes that the interaction of family members will gradually become routine in nature” and “13. The family desires peace for their family members” was enhanced immediately after the FLR. The wife commented, “*We really go about our daily lives treating our family like air. We take that for granted, but now we are grateful for it*,” and the husband agreed. The wife said, “*Until now I was terrified, but now I feel a sense of security. I have greater peace of mind*,” and she was once again aware of the joy she felt that her husband’s condition had stabilized and that he could continue receiving care at home.

#### 3.2.5. Changes in Family Spirituality in Family ID 5

Changes in family spirituality are indicated in Table 8. In the family internal environment system, belief in “1. The family believes that the family exists for the sake of children and grandchildren” was enhanced immediately after the FLR. The second daughter said, “*We leave descendants because we have ancestors. I think that family members are important even if they are far apart because of this connection. I think that we would not exist if even one person was missing*,” reinforcing the belief that the family is connected to the next generation. In addition, belief in “2. The family believes that the wishes of family members should be respected” was enhanced immediately after the FLR. The second daughter said, “*We should not forget about respect. We need to think about our father in the past, not just our feeble father today*” and “*I looked back on things I had forgotten and reconsidered this (PFS), and there are some things I have noticed again. I also want to do something to make my father’s life easier*.” The second daughter recalled that her father was still the same father she remembered; her feelings of gratitude and respect for him became stronger, and she showed a willingness to change her attitude toward caregiving.

In the family system unit, belief in “6. The family believes that family members should support each other,” “8. The family believes that the family is irreplaceable,” and “9. The family believes that the family has a trusting relationship that allows family members to talk to each other” was enhanced immediately after the FLR, and that in “11. The family believes that the family is a safe place” was maintained immediately after the FLR. The second daughter said, “*I think a family works by working together and supporting each other*,” “*My siblings say kind things to me. If they didn’t have those, I would really be heartbroken, but they are considerate of me*,” and “*Even when they are far away, they care about me and support me. They’re not close by and can’t do anything, but everyone adores our father*,” and the wife agreed. Family members were concerned about each other; although they were unable to provide physical care, they were supported emotionally, and they reaffirmed that there were family members who could sympathize with their anxieties and distress regarding caregiving. In addition, belief in “7. The family believes that the family is a kind of microcosm of society” was enhanced one month after the FLR. The second daughter said, “*I had never looked back like this before, but now I can see things from a broader perspective. Not just those close to me (my children and a grandchild), but also my older sister and younger brother*.” Her thoughts were strengthened as she thought deeply about her family through the FLR.

Belief in the chrono system’s “12. The family believes that the interaction of family members will gradually become routine in nature,” “13. The family desires peace for their family members,” and “14. The family believes that family members should share time together” was enhanced immediately after the FLR. The second daughter said, “*I want to live a happy life rather than fighting, so I try to interact as peacefully as possible. I also want to have peace of mind*.” “*By doing FLR, I was able to think about the hardships my father faced in the past and that time in his life. He was someone who worked hard and thought about his family, so I have to treat him right*.” The wife agreed. She was reminded of the effectiveness of being able to care for her husband and maintain his life at home, and her motivation for her future caregiving life was increased.

#### 3.2.6. Changes in Family Spirituality in Family ID 6

Changes in family spirituality are indicated in Table 9. Belief in family system unit items “6. The family believes that family members should support each other,” “8. The family believes that the family is irreplaceable,” “10. The family believes that family members should be close to each other,” and “11. The family believes that the family is a safe place” was enhanced immediately after the FLR. The eldest daughter said, “*I talk about how I wish we could do it this way and try to ease my mother’s mental state a little. I feel like we can overcome this by sharing information as we have done in the past*.” “*First, we have to accept the current situation. Then, if we can do this once, we can do this again. Even though I feel depressed, I think positively*.” “*I want to do my best until I can take care of my father*.” The mother agreed. By supporting each other as a family, so that the father could continue living at home, the family was able to regain their goals and have a positive attitude toward living at home. Furthermore, one month after the FLR, the wife and eldest daughter commented, “*I feel like I’ve finally been able to put it into words*,” and “*It made me realize just how little I have thought about the significance of my family’s existence. I need to be grateful*.” As a result, the family spirituality, which was strengthened immediately after the FLR, was further enhanced one month after the FLR.

Belief in the chrono system’s “14. The family believes that family members should share time together” was enhanced immediately after the FLR. The eldest daughter commented, “*Because family members exist, there is a significance in being together*.” The wife commented, “*I always think about my husband. If something were to happen, I wonder if I could endure it. They can’t do anything for me, but I feel a sense of satisfaction just by being there*,” and she reaffirmed the value of time spent with family members.

### 3.3. Appraisals of Using the PFS in the FLR

Regarding the PFS, participants said, “*When we fight, we look back at this* (ID 1 and 5),” “*When I’m feeling down, I want to look back at this* (ID 5),” “*I want to keep it within reach so I can look back at it whenever I want* (ID 4),” “*I want to share it with other family members* (ID 2, 3, 4, and 5),” “*It sparked conversations with family members* (ID 3),” “*I want to share it with family members and enjoy the different memories and different perspectives on memories that emerge* (ID 5),” “*It was good that the family history was summarized so that it could be seen and understood* (ID 3, 4, 5, and 6),” and “*When a family member passes away, I want to use it for the funeral story* (ID 5).”

## 4. Discussion

### 4.1. Changes in Family Spirituality Through the FLR

#### 4.1.1. Family Spirituality in the Family Internal Environment System

O’Hora et al. explain that the FLR allows family members to share their stories and experiences, which enables them to deepen their understanding of the stories of other family members [11]. In those families with IDs 1 and 5, the family members reminisced about their childhoods and parents, realizing the hardships and love of their parents at the time and renewing their respect for them. They then gave meaning to the family’s experiences and realized that their parents’ hardships and love were dedicated to their children and grandchildren. Furthermore, they now recalled how they themselves had thought of and devoted themselves to their children. In this process, the belief that “1. The family believes that the family exists for the sake of children and grandchildren” was enhanced.

Life reviews targeting bereaved families show increased feelings of gratitude, the urge to express gratitude to others, and a desire to repay the kindness shown to them [17]. Families with IDs 1 and 5 reinforced their feelings of gratitude to family members, showed respect to them, expressed a desire to be closer to them through the FLR, and reconsidered how they interacted with their family members. This reaffirmed and enhanced the importance of “2. The family believes that the wishes of family members should be respected” for the family.

Regarding “5. The family believes that it is important for family members to trust spirituality,” research exists that indicates it is necessary to provide family interventions that correspond to the family’s spiritual beliefs and values [5], and Family ID 1 reinforced their desire to respect the things and values that family members hold dear, which we believe was enhanced.

#### 4.1.2. Family Spirituality of the Family System Unit

All the families in this study suffered spiritually in the family system unit. These are core family beliefs that emphasize the physical and psychological bonds of the family [15], such as “6. The family believes that family members should support each other”; “8. The family believes that the family is irreplaceable”; “9. The family believes that the family has a trusting relationship that allows family members to talk to each other”; and “11. The family believes that the family is a safe place.” In order to exist as a family, it is necessary for a family to establish an identity based on family ties [18], and we believe that such an identity reflects the meaning of the family’s existence. Therefore, we believe that all the families in this study suffered spiritually in the family system unit.

An individual’s life review reassures them that they are alive and that they are the main actors in their own lives [19]. Although the families with IDs 1, 3, and 5 recognized the importance of “6. The family believes that family members should support each other,” in reality, they had difficulty in realizing that family members were supporting each other, and that their family spirituality was unstable. However, through the FLR, they realized that in the course of their everyday lives, family members had cooperated and helped each other when they encountered trouble. This enabled them to recognize that “6. The family believes that family members should support each other” was the main actor in their family lives, and it is believed that this could be maintained and/or enhanced.

In the FLR, members may find themselves in a system and become strongly aware that their family exists within, and is connected to, various systems [11]. For Family ID 1, the FLR led them to believe that the social situation affects the family and establishes the meaning of the family’s existence, and we believe that this conferred importance to “7. The family believes that the family is a kind of microcosm of society.” For Family ID 5, the FLR helped them to broaden their perspective, not only toward their own family but also to the surrounding environment with which the family is closely related. Furthermore, by gathering together as family members, they were able to view the family from a bird’s-eye perspective, and we believe that one month after the FLR, belief in “7. The family believes that the family is a kind of microcosm of society” was enhanced.

An individual’s life review has the effect of increasing an individual’s self-esteem, and some people realize that their own life is not mundane but is instead irreplaceable [19]. Similarly, in the FLR, we believe that the participants reinforced their belief that family is not something to be taken for granted but is an irreplaceable entity, and realized the meaning of the existence of their unique family. In the process, we believe that “8. The family believes that the family is irreplaceable” was enhanced and/or maintained.

Personal life reviews improve the participants’ sense of being respected and understood [20]. Family ID 5 felt that sharing their family story in the FLR made them realize that their thoughts and feelings were understood by other family members and believed that this enhanced their belief in “9. The family believes that the family has a trusting relationship that allows family members to talk to each other.” The FLR encourages family members to cooperate in solving family problems [11]. Family ID 3 reconsidered their relationship with their eldest daughter and became more actively involved in resolving family problems through the FLR. This promoted communication between family members, and we believe that one month after the FLR, belief in “9. The family believes that the family has a trusting relationship that allows family members to talk to each other” was enhanced.

FLR offers a new perspective on family stories [11]. Family ID 6 found that time spent together was a valuable experience for the family, and reaffirmed during the FLR that continuing home care is a top priority for the family and that they desire to feel reassured by having family members nearby. Belief in “10. The family believes that family members should be close to each other” refers to family spirituality, which allows family members to feel a sense of interconnection, and we determined that reaffirming this awareness enhanced this belief.

By participating in the FLR, families confirm and strengthen their current family relationships [11]. The families with IDs 1, 4, 5, and 6 strengthened their belief that their family would accept them, whatever the situation, and felt a stronger psychological bond with their family. This is thought to have enhanced and/or maintained belief in “11. The family believes that the family is a safe place.”

#### 4.1.3. Family Spirituality in the Chrono System

Life reviews with bereaved families show increased feelings of gratitude, which they express to others, along with a desire to reciprocate the kindness they have received [17]. Through the FLR, families are able to appreciate their family members and the value of their family, gradually reaffirming the importance of the presence of family members, something that they had normally taken for granted. From this, we believe that belief in “12. The family believes that the interaction of family members will gradually become routine in nature” was enhanced.

By conducting an individual life review, subjects become grateful for their lives, accept reality, and become future-oriented [21]. Families with members receiving care at home strongly hoped that their family would live peacefully and that the family would continue to exist, even in the midst of an uncertain caring situation; in the process, it was believed that, as stated in “13. The family desires peace for their family members,” the family’s hope for the future was enhanced.

Families place importance on the time that family members spend together, and the FLR provides an opportunity to confirm and enjoy the history that the family has built up through their time together [11]. We believe that the FLR helps family members realize that the family’s history itself constitutes a value of the family’s existence, underscoring the statement in “14. The family believes that family members should share time together.”

FLR promotes intergenerational ties, such as by enabling the continuity of family members’ stories between previous and future generations [11]. Family ID 3 realized that through the FLR, the mother and eldest daughter had inherited the fields that their father had treasured, and that they hoped that their grandchildren would inherit them in the future. This helped them realize that their family story was connected to the next generation. As a result, the family became conscious of the subconscious belief that protecting the fields was the family’s mission, and they recognized that “15. The family believes that they should bequeath their own culture” was an element in the family’s reason for being.

### 4.2. The Significance of Using the PFS in the FLR and Its Implications for Family Nursing

In a study that used a life review book (album) in which subjects pasted stories and photos of their personal life reviews, this book became a personal legacy to convey their values in life and was a means to be remembered by loved ones [22]. Similarly, in this study, the PFS was useful for confirming important family experiences and their meanings, and for conveying family history to other family members, making for a family legacy that allowed them to relive past events. O’Hora et al. [11] described the importance of sharing legacies in the FLR, reporting that by sharing memories between and across generations, looking back on the past, and remembering each other as something that will be passed down by members of the family, they became more aware of the family’s value. From this, it is believed that sharing the PFS as a legacy within the family can increase awareness of the value of the family’s existence.

Sytsma et al. [23] reported that a spiritual legacy document (SLD), which included the subjects’ words and photos, was used in individual life reviews, and that the effect was that individuals were able to review their own lives and recognize how they had overcome challenges in life. The PFS can be visualized as a table that indicates the family’s perception of family events and the changes in the family caused by family events. This allows the family to confirm how they overcame family events and realized their strengths, which is thought to have led to a positive perception of the meaning of the family’s existence and promoted changes in family spirituality. Since the FLR using the PFS can help family members to recognize the meaning of their existence and strengthen family spirituality, it is thought that it can also be used in family nursing for families undergoing spiritual suffering.

### 4.3. Limitations of the Study

This study recruited families at two visiting nurse stations in a city in Japan and targeted a small number of families that included older adults, thereby limiting the generalizability of the study results. Future studies targeting a variety of families will be necessary to verify the usefulness of the FLR using the PFS. In addition, this study followed a before-and-after comparative design in the intervention group, and factors that may have influenced changes in family spirituality apart from the FLR cannot be ruled out. Future studies should be conducted using a research design that includes a control group.

## 5. Conclusions

We investigated the effects of family intervention through the FLR, using the PFS on changes in family spirituality in families undergoing spiritual suffering. The effects of the FLR were evaluated using a mixed-methods approach, using a semi-structured interview regarding family spirituality and scores from the Family Spirituality Index. The FLR can maintain, enhance, confer, and actualize family spirituality as applied to the spiritual suffering of the family, demonstrating its usefulness as a method of supporting family spirituality. In so doing, the PFS became a legacy for the family and raised awareness in family members of the value of the family’s existence.

## Figures and Tables

**Table 1 nursrep-15-00277-t001:** Examples of questions used in the FLR.

**First Session**
I would like to ask you in chronological order, working from the past. Among the memories shared by your family, what is the first event that comes to mind that left an impression on your family or that you think was important to your family? When and how did that event happen? What change occurred in your family as a result of that event? When and how did that change occur? Why do you think that change occurred? What does your family think about that change? Why do you think that the change happened that way? Do you think the impact that the change had on your family at that time was positive or negative? What meaning did that experience have for your family?
**Second Session**
From what you said, and looking at your family timeline, what kind of growth has your family experienced? Based on what you have said, what is most important in your family life right now? Also, why is that? Based on what you have said, what do you think is most valuable to your family (your family motto)? Why do you think so? What do you take pride in about your family? Based on what you have said, and looking at your family timeline, what kind of family do you think your family is, overall? Why do you think so?

**Table 2 nursrep-15-00277-t002:** Contents of the Family Spirituality Index.

System	Items in the Family Spirituality Index
Family internal environment system	1. The family believes that the family exists for the sake of children and grandchildren 2. The family believes that the wishes of family members should be respected 3. The family believes that parents do not want to be looked after by children 4. The family believes that the more family members there are, the more likely it is that they will be happier 5. The family believes that it is important for family members to trust spirituality
Family system unit	6. The family believes that family members should support each other 7. The family believes that the family is a kind of microcosm of society 8. The family believes that the family is irreplaceable 9. The family believes that the family has a trusting relationship that allows family members to talk to each other 10. The family believes that family members should be close to each other 11. The family believes that the family is a safe place
Chrono system	12. The family believes that the interactions of family members will gradually become routine in nature 13. The family desires peace for their family members 14. The family believes that family members should spend shared time together 15. The family believes that they should bequeath their own culture
Micro system	16. The family believes that relationships between its members differ from those of people outside the family 17. The family believes that it is important to have connections with those close to it
Macro system	18. The family believes that mutual understanding with medical professionals is necessary
Supra system	19. The family believes in religious teachings 20. The family believes that religion is a blessing 21. The family believes that family spirituality and religion can sometimes overlap

**Table 3 nursrep-15-00277-t003:** Attributes of the target family and family spiritual suffering.

ID	Family Overview		Contents of Family Spiritual Suffering
1	A family of five, consisting of the husband (in his 80s), the wife (in her 80s), the eldest son (in his 50s), the eldest daughter (in her 50s), and her child (in his 20s). The husband and wife live together, the eldest son is in a facility for people with disabilities, and the eldest daughter and her child live alone. The husband is in need of care, and the wife and eldest daughter work together to provide him with home care. The wife and eldest daughter participated in the family life review at all three points.		The eldest daughter explained that although she wanted to respect her parents’ wishes, she sometimes felt irritation and self-loathing when she was with them, undergoing family spiritual suffering in the family’s internal environment system. In addition, the eldest daughter is anxious about losing family members and not knowing when she will be unable to see them again, as her parents are getting older, her brother is in a facility, visitation restrictions are becoming stricter, and she is also undergoing family spiritual suffering in the family system unit.
2	A family of six, consisting of the mother (in her 80s), the eldest son (in his 50s), the second son (in his 50s), the third son (in his 50s), the eldest daughter (in her 50s), and the second daughter (in her 50s). The mother lives alone, but her children work together to provide her with home care. The mother, eldest son, and eldest daughter participated in the family life review at all three points.		Since the mother became dependent on care, family members are worried about how the family should devise ways to support their mother’s life while she is recovering. The family is experiencing family spiritual suffering in the family system unit.
3	A family of four, consisting of the mother (in her 70s), the eldest daughter (in her 40s), the eldest daughter’s husband (in his 40s), and the eldest daughter’s child (in her teens). All family members live together. The mother and eldest daughter participated in the family life review at all three points.		Since the death of the father, communication in the family has decreased, and the ties between family members have become weaker; the mother and the eldest daughter are concerned that the relationship between the eldest daughter and her husband is deteriorating, and they are anxious about how the family will continue in the future. The family is undergoing family spiritual suffering in the family system unit and the chrono system.
4	A family of four, consisting of the husband (in his 80s), the wife (in her 80s), the eldest son (in his 50s), and the second son (in his 50s). The husband and wife live together, while the eldest son and second son live far away. The husband is receiving care at home. The husband and wife participated in the family life review at all three points.		Although the husband is currently receiving stable care at home, anxiety is felt about not knowing when his condition will change and about the burden of caring for him, and the family is suffering spiritually in the family system unit and chrono system.
5	A family of nine, consisting of the husband (in his 80s), the wife (in her 80s), the eldest daughter (in her 50s) and her child (in her teens), the second daughter (in her 50s) and her child (in his 30s), and the second daughter’s other child and her husband (in their 30s and 30s) and their daughter (under 5 years old). The husband and wife live together, and the second daughter and her child live next door. The husband is currently receiving home care. The wife and second daughter participated in the family life review at all three points.		The husband is continuing to receive care at home, but as time passes, the burden of caring for him is increasing. Due to dementia, the husband is losing the attributes of fatherhood that the family expects him to have. The family is feeling distressed by the reality that their image of their father is changing, and the family is suffering spiritually in the family internal environment system, family system unit, and chrono system.
6	A family of seven, consisting of the husband (in his 80s), wife (in her 80s), eldest daughter (in her 50s), the eldest daughter’s husband (in his 50s), the eldest daughter’s two children (one in her 20s and the other in his teens), and the second daughter (in her 50s). The husband and wife live together, and the eldest daughter’s family lives nearby. The husband is currently hospitalized. The wife and eldest daughter participated in the family life review at all three points.		The family’s goal is for the husband and wife to live together at home, but this can no longer be achieved due to the husband’s hospitalization. In addition, the family is losing sight of their family goal because it is unclear whether the father will be able to return to living at home after being discharged from the hospital, and the family is suffering spiritually in the family system unit and the chrono system.

**Table 4 nursrep-15-00277-t004:** Changes in family spirituality for Family ID 1.

System	Family Spirituality Index Items (Numbers in Table 2)	Family Spirituality Index Scores			Changes in Family Spirituality (Assessed Using a Mixed Quantitative and Qualitative Method)	
		Just Before FRL	Immediately After FRL	One Month After FLR	Immediately After FRL	One Month After FLR
Family internal environment system	1	7	10	10	enh	enh
	2	5	7	8	enh	enh
	3	5	7	7		
	4	5	5	5		
	5	8	8	10		enh
Family system unit	6	10	10	10	main	main
	7	5	8	10	conf	conf
	8	10	10	10	main	enh
	9	10	10	10		
	10	8	8	8		
	11	10	10	10	enh	

Note: FRL = family life review, enh = enhancement of family spirituality, main = maintenance of family spirituality, conf = conferment of family spirituality.

**Table 5 nursrep-15-00277-t005:** Changes in family spirituality for Family ID 2.

System	Family Spirituality Index Items (Numbers in Table 2)	Family Spirituality Index Scores			Changes in Family Spirituality (Assessed Using a Mixed Quantitative and Qualitative Method)	
		Just Before FRL	Immediately After FRL	One Month After FLR	Immediately After FRL	One Month After FLR
Family system unit	6	8	8	8		
	7	8	8	8		
	8	10	10	10	main	main
	9	8	8	8		
	10	7	7	7		
	11	8	8	8		

Note: FRL = family life review, main = maintenance of family spirituality.

**Table 6 nursrep-15-00277-t006:** Changes in family spirituality for Family ID 3.

System	Family Spirituality Index Items (Numbers in Table 2)	Family Spirituality Index Scores			Changes in Family Spirituality (Assessed Using a Mixed Quantitative and Qualitative Method)	
		Just Before FRL	Immediately After FRL	One Month After FLR	Immediately After FRL	One Month After FLR
Family system unit	6	10	10	10	main	main
	7	8	7	7		
	8	10	10	10		
	9	2	2	3		enh
	10	5	5	5		
	11	10	10	10		
Chrono system	12	2	2	2		
	13	10	10	10		
	14	5	5	5		
	15	2	4	4	act	act

Note: FRL = family life review, main = maintenance of family spirituality, enh = enhancement of family spirituality, act = actualization of family spirituality.

**Table 7 nursrep-15-00277-t007:** Changes in family spirituality for Family ID 4.

System	Family Spirituality Index Items (Numbers in Table 2)	Family Spirituality Index Scores			Changes in Family Spirituality (Assessed Using a Mixed Quantitative and Qualitative Method)	
		Just Before FRL	Immediately After FRL	One Month After FLR	Immediately After FRL	One Month After FLR
Family system unit	6	10	10	10+	enh	enh
	7	8	8	8		
	8	10	10	10+	enh	enh
	9	10	10	10+	enh	enh
	10	5	5	5		
	11	10	10	10+	enh	enh
Chrono system	12	10	10	10+	enh	enh
	13	10	10	10+	enh	enh
	14	6	7	7		
	15	8	5	5		

Note: FRL = family life review, enh = enhancement of family spirituality.

**Table 8 nursrep-15-00277-t008:** Changes in family spirituality for Family ID 5.

System	Family Spirituality Index Items (Numbers in Table 2)	Family Spirituality Index Scores			Changes in Family Spirituality (Assessed Using a Mixed Quantitative and Qualitative Method)	
		Just Before FRL	Immediately After FRL	One Month After FLR	Immediately After FRL	One Month After FLR
Family internal environment system	1	10	10+	10+	enh	enh
	2	10	10+	10+	enh	enh
	3	5	5	5		
	4	9	8	8		
	5	7	7	7		
Family system unit	6	10	10+	10+	enh	enh
	7	8	8	10		enh
	8	10	10+	10+	enh	enh
	9	10	10+	10+	enh	enh
	10	8	8	9		
	11	10	10	10+	main	main
Chrono system	12	6	9	10	enh	enh
	13	10	10+	10+	enh	enh
	14	6	8	8	enh	enh
	15	7	7	7		

Note: FRL = family life review, enh = enhancement of family spirituality, main = maintenance of family spirituality.

**Table 9 nursrep-15-00277-t009:** Changes in family spirituality for Family ID 6.

System	Family Spirituality Index Items (Numbers in Table 2)	Family Spirituality Index Scores			Changes in Family Spirituality (Assessed Using a Mixed Quantitative and Qualitative Method)	
		Just Before FRL	Immediately After FRL	One Month After FLR	Immediately After FRL	One Month After FLR
Family system unit	6	5	6	9	enh	enh
	7	5	5	6		
	8	9	9	9	enh	enh
	9	9	9	9		
	10	6	7	8	enh	enh
	11	7	8	8	enh	enh
Chrono system	12	8	8	9		
	13	9	9	9		
	14	4	5	6	enh	enh
	15	5	5	6		

Note: FRL = family life review, enh = enhancement of family spirituality.

## Data Availability

The datasets used and analyzed during the current study are available from the corresponding author on reasonable request.

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
