# Peer review of "Changes in Family Spirituality in Response to Family Intervention Utilizing the Family Life Review"

_nursrep, 2025, doi:10.3390/nursrep15080277_

Round 1
Reviewer 1 Report
Comments and Suggestions for Authors
As a reviewer I have no conflicts of interest with this work.
Detailed Article Report
Article Title: The Effects of Family Intervention Utilizing Family Life Review on Changes in Family Spiritual Suffering
Authors: Naohiro Hohashi and Haruka Yano
Brief summary
The study investigates the impact of a family intervention using family life review (FLR) on changes in family spirituality in families experiencing spiritual suffering, through a quasi-experimental study. The results show that FLR can maintain, enhance, confer and update family spirituality, becoming a legacy for the family and increasing awareness of the value of its existence. As strengths, the paper addressed an underexplored topic, it is based on an adequate nursing theoretical model or uses triangulation and discussion between evaluators to reduce the risk of bias in interviews.
General concept comments
- The main inadequacy at the level of expression and methodology is the use of the concept of "effectiveness". In fact, one objective is “to verify its effectiveness as family spirituality support” and conclusions state "FLR can... demonstrating its effectiveness as a method of supporting family spirituality." It is not appropriate to claim that an intervention is effective when robust statistical analysis (including p-values) have not been conducted to validly establish this affirmation. In a quasi-experimental study such as the one presented with a pre-test post-test descriptive analysis, although it may be a good starting point, it is not enough to verify the effectiveness of an intervention.
A descriptive analysis (such as comparing pre-test and post-test means) can show trends, but it does not control for external or confounding variables (which has been mentioned in limitations), does not allow determining whether the changes are statistically significant, and does not estimate the size of the effect or its accuracy.
Based on the above, it is recommended to remove from the paper the references to the analysis and interpretation of the effectiveness of the intervention, indicating only the descriptive changes observed. This can be taken as a starting point (prospective) for analyse effectiveness in a larger sample and with other types of statistical analysis.
- Referring the Family Spirituality Index, it is affirmed that “was distributed to family members after the semi-structured interview and collected after they had completed it.”. At least two members of each family participated in the family life review, but only one punctuation of the index was presented in tables 4-9 for each timepoint (before FLR, after FLR, one month after FLR). Then, was the punctuation the mean obtained of the members of each family? There was used the lowest/highest punctuation? Was the presented punctuation established by the family nursing researchers? Only one member completed the index? There was the same respondent at the three measurements? It should be clarified how the score presented in the tables was obtained.
- At “Ethical considerations” the Anonymization and pseudonymisation of data should be established (how it has been transformed so that it cannot be easily linked to identifiable individuals).
- The method indicates that the participating families were recruited from two home nursing stations that cooperated with the study, but the city or country in which it was carried out is not indicated at any point in the study. Given the profile of the authors and the funding, it is highly likely that the study was carried out in Japan, but this fact must be clearly indicated in the method.
- Point "3.3. Effects of Using PFS in FLR" is too succinct. According to the methodology "FLR was conducted using the Plot of Family Story (PFS), a tool for reviewing family history based on the Concentric Sphere Family Environment Theory", representing therefore an important aspect in the development of the study. The PFS is described in lines 74-89 and is used in the first and second sessions. However, only the appraisals of the families regarding the instrument are described in point 3.3. It would be advisable to include more descriptive details such as the number of positive and negative events in families or what aspects these two types of events were related to, for example: births, moves, illnesses, deaths...
- Regarding the references, only 4 (19%) correspond to publications from the last 5 years, and 10 (47.6%) from the last 10 years. Four are self-citations (19%). All this can be understandable because it is a subject that has been little studied, more specifically in the way addressed here. In the introduction, it is made clear that the work addresses family spirituality based on the theoretical bases of two previous publications by the first author of this work.
Despite this, it is missing in the introduction (and also in the discussion, that presents few new references with respect to those in the introduction) to name other existing previous works that have addressed family spirituality or family review intervention, such as:
- Taylor, S. D., Stahl, M., & Distelberg, B. (2021). The Spiritual Perspective Scale–Family Version (SPS-FV): A Tool for Assessing Perceptions of Spirituality Within Families. Journal of Family & Consumer Sciences, 113(2), 50–61. https://doi.org/10.14307/JFCS113.2.50
- Islamia, I., & Marliani, R. (2023). Exploring family strength: Vulnerability factors and the role of spirituality in family resilience during COVID-19 pandemic. Anfusina: Journal of Psychology, 6(1). https://doi.org/10.24042/00202362048300
- Chou, F., Black, T., Huang, C., Tran, A., Yan, M., & Boothroyd, S. (2024). The development of a dyadic family life review intervention for the Asian diaspora: A practice article. Journal of Marital and Family Therapy, 51(1). https://doi.org/10.1111/jmft.12750
- Sari, N. A., Widyastuti, M., & Rifah, P. A. (2021). Spirituality and anxiety in critical care patients' families: A systematic review. Jurnal Ilmu dan Teknologi Kesehatan, 9(1), 1–12. https://doi.org/10.32668/jitek.v9i1.552
- Hodge, D. R. (2000). Spiritual ecomaps: A new diagrammatic tool for assessing marital and family spirituality. Journal of Marital and Family Therapy, 26(2), 217–228. https://doi.org/10.1111/j.1752-0606.2000.tb00291.x
- Kim, S.-S., Kim-Godwin, Y. S., & Koenig, H. G. (2016). Family spirituality and family health among Korean-American elderly couples. Journal of Religion and Health, 55(2), 729–746. https://doi.org/10.1007/s10943-015-0107-5
Naming the most relevant results/conclusions of these publications, indicating the similarities, differences, and most important contributions to the work presented here, is ethically desirable when recognizing the work of other researchers. In addition, the novel contribution of the perspective and results of this study is better recognized. I recommend that authors consider including in the introduction and/or discussion sections some of the works indicated, in the terms mentioned above.
- The manuscript is clear and presented in a well-structured manner. The results are reproducible based on the details given in the methods section. The tables are appropriate and easy to understand. The ethics statements are adequate.
Author Response
Comments 1: The main inadequacy at the level of expression and methodology is the use of the concept of "effectiveness". In fact, one objective is “to verify its effectiveness as family spirituality support” and conclusions state "FLR can... demonstrating its effectiveness as a method of supporting family spirituality." It is not appropriate to claim that an intervention is effective when robust statistical analysis (including p-values) have not been conducted to validly establish this affirmation. In a quasi-experimental study such as the one presented with a pre-test post-test descriptive analysis, although it may be a good starting point, it is not enough to verify the effectiveness of an intervention.
A descriptive analysis (such as comparing pre-test and post-test means) can show trends, but it does not control for external or confounding variables (which has been mentioned in limitations), does not allow determining whether the changes are statistically significant, and does not estimate the size of the effect or its accuracy.
Based on the above, it is recommended to remove from the paper the references to the analysis and interpretation of the effectiveness of the intervention, indicating only the descriptive changes observed. This can be taken as a starting point (prospective) for analyse effectiveness in a larger sample and with other types of statistical analysis.
Response 1: The word "effectiveness" has been discontinued in the paper (from lines 95, 157, 158, and 519).
Comments 2: Referring the Family Spirituality Index, it is affirmed that “was distributed to family members after the semi-structured interview and collected after they had completed it.”. At least two members of each family participated in the family life review, but only one punctuation of the index was presented in tables 4-9 for each timepoint (before FLR, after FLR, one month after FLR). Then, was the punctuation the mean obtained of the members of each family? There was used the lowest/highest punctuation? Was the presented punctuation established by the family nursing researchers? Only one member completed the index? There was the same respondent at the three measurements? It should be clarified how the score presented in the tables was obtained.
Response 2: The following has been added from lines 190 to 191.
All participating family members consulted and agreed upon the questionnaire before completing one Family Spirituality Index.
In Table 3, we have indicated that the same family members participated in the family life review at all three points.
Comments 3: At “Ethical considerations” the Anonymization and pseudonymisation of data should be established (how it has been transformed so that it cannot be easily linked to identifiable individuals).
Response 3: The following has been added from line 215 to line 218.
The information on the target families was anonymised by ID number using a correspondence table with names to ensure that families and individuals could not be identified. The anonymised data and the correspondence table were stored securely in separate lockable storage cabinets to prevent information leaks.
Comments 4: The method indicates that the participating families were recruited from two home nursing stations that cooperated with the study, but the city or country in which it was carried out is not indicated at any point in the study. Given the profile of the authors and the funding, it is highly likely that the study was carried out in Japan, but this fact must be clearly indicated in the method.
Response 4: The following has been added on line 142.
The target families were recruited at a city in Japan from two visiting nurse stations that cooperated with the study,.
Comments 5: Point "3.3. Effects of Using PFS in FLR" is too succinct. According to the methodology "FLR was conducted using the Plot of Family Story (PFS), a tool for reviewing family history based on the Concentric Sphere Family Environment Theory", representing therefore an important aspect in the development of the study. The PFS is described in lines 74-89 and is used in the first and second sessions. However, only the appraisals of the families regarding the instrument are described in point 3.3. It would be advisable to include more descriptive details such as the number of positive and negative events in families or what aspects these two types of events were related to, for example: births, moves, illnesses, deaths...
Response 5: "3.3. Effects of Using PFS in FLR" summarizes the impressions of families about the benefits of FLR, so the title will be changed to "3.3. Appraisals of Using PFS in FLR". Details of these events are described in "3.1. Overview of the Target Families" and "3.2. Changes in Family Spirituality for Each Family."
Comments 6: Regarding the references, only 4 (19%) correspond to publications from the last 5 years, and 10 (47.6%) from the last 10 years. Four are self-citations (19%). All this can be understandable because it is a subject that has been little studied, more specifically in the way addressed here. In the introduction, it is made clear that the work addresses family spirituality based on the theoretical bases of two previous publications by the first author of this work.
Despite this, it is missing in the introduction (and also in the discussion, that presents few new references with respect to those in the introduction) to name other existing previous works that have addressed family spirituality or family review intervention, such as:
Taylor, S. D., Stahl, M., & Distelberg, B. (2021). The Spiritual Perspective Scale–Family Version (SPS-FV): A Tool for Assessing Perceptions of Spirituality Within Families. Journal of Family & Consumer Sciences, 113(2), 50–61. https://doi.org/10.14307/JFCS113.2.50
Islamia, I., & Marliani, R. (2023). Exploring family strength: Vulnerability factors and the role of spirituality in family resilience during COVID-19 pandemic. Anfusina: Journal of Psychology, 6(1). https://doi.org/10.24042/00202362048300
Chou, F., Black, T., Huang, C., Tran, A., Yan, M., & Boothroyd, S. (2024). The development of a dyadic family life review intervention for the Asian diaspora: A practice article. Journal of Marital and Family Therapy, 51(1). https://doi.org/10.1111/jmft.12750
Sari, N. A., Widyastuti, M., & Rifah, P. A. (2021). Spirituality and anxiety in critical care patients' families: A systematic review. Jurnal Ilmu dan Teknologi Kesehatan, 9(1), 1–12. https://doi.org/10.32668/jitek.v9i1.552
Hodge, D. R. (2000). Spiritual ecomaps: A new diagrammatic tool for assessing marital and family spirituality. Journal of Marital and Family Therapy, 26(2), 217–228. https://doi.org/10.1111/j.1752-0606.2000.tb00291.x
Kim, S.-S., Kim-Godwin, Y. S., & Koenig, H. G. (2016). Family spirituality and family health among Korean-American elderly couples. Journal of Religion and Health, 55(2), 729–746. https://doi.org/10.1007/s10943-015-0107-5
Naming the most relevant results/conclusions of these publications, indicating the similarities, differences, and most important contributions to the work presented here, is ethically desirable when recognizing the work of other researchers. In addition, the novel contribution of the perspective and results of this study is better recognized. I recommend that authors consider including in the introduction and/or discussion sections some of the works indicated, in the terms mentioned above.
The manuscript is clear and presented in a well-structured manner. The results are reproducible based on the details given in the methods section. The tables are appropriate and easy to understand. The ethics statements are adequate.
Response 6: We have read the six papers you presented. We have cited two papers from lines 45 to 48. We have also cited one paper from lines 395 to 396.
Reviewer 2 Report
Comments and Suggestions for Authors
Dear authors,
I find this an interesting study because the development of family spirituality has a great influence on the health of its members.
The introduction broadly addresses the research topic and allows the unfamiliar reader to delve into family spirituality and the interventions to improve it.
The objective of the study needs to be clarified. After reading the manuscript, I believe that the aim is to improve family spirituality through interventions, since at no point is the degree of family suffering assessed quantitatively or qualitatively; it is impossible to know if it has changed with the proposed measures.
The methodology used allows the authors to satisfactorily address the study problem and achieve the proposed objective, although the type of sampling used should be specified.
The reason for choosing a specific week and month between measurements should also be indicated.
The results tables report the changes in family spirituality observed between measurements, indicating that the results were evaluated using a mixed quantitative and qualitative method not mentioned in the methodology. The nature of this method should be specified.
Was any statistical test used to assess the quantitative differences between the two Family Spirituality Index measurements? How were qualitative changes assessed?
The results are presented in detail and in tables that facilitate understanding, although it should be clarified why not all items of the Family Spirituality Index appear in the tables. Why were the ones that did appear chosen? Why do items 17 through 21 not appear in any of the tables?
The discussion qualitatively analyzes the results obtained and establishes relationships with previous studies; although at no point does it perform a quantitative analysis specifying which results obtained after the intervention was statistically significant. If the results were assessed quantitatively, as stated in the tables, the significance of the changes produced by the intervention should be indicated.
I believe that the limitations should indicate those arising from the type of sample used.
I imagine the conclusions are consistent with the results obtained, although to do so I would need to know the method used to assess the results and the statistical significance of the quantitative changes that occurred in the Family Spirituality Index after the intervention.
The references are adequate and mostly up-to-date, allowing the reader to delve deeper into the research question.
Kind regards.
Author Response
Comments 1: I find this an interesting study because the development of family spirituality has a great influence on the health of its members.
The introduction broadly addresses the research topic and allows the unfamiliar reader to delve into family spirituality and the interventions to improve it.
The objective of the study needs to be clarified. After reading the manuscript, I believe that the aim is to improve family spirituality through interventions, since at no point is the degree of family suffering assessed quantitatively or qualitatively; it is impossible to know if it has changed with the proposed measures.
Response 1: The title of the paper has been changed to "Changes in Family Spirituality in Response to Family Intervention Utilizing Family Life Review."
In lines 93 to 96, objective has been changed to the following.
Based on the above, this study targeted families undergoing family spiritual suffering and aimed to clarify the ways in which family spirituality changes before and after family intervention through FLR using PFS, and to verify its improvement as family spirituality support.
Comments 2: The methodology used allows the authors to satisfactorily address the study problem and achieve the proposed objective, although the type of sampling used should be specified.
Response 2: In lines 141 to 145, we added to the effect that it was purposive sampling as per below.
The target families were recruited at a city in Japan from two visiting nurse stations that cooperated with the study, and from among the families expressing a willingness to participate in the study, those who the researchers judged to be experiencing family spiritual suffering were recruited using purposive sampling.
Comments 3: The reason for choosing a specific week and month between measurements should also be indicated.
Response 3: This is already stated in lines 148 to 150 and lines 161 to 168.
Comments 4: The results tables report the changes in family spirituality observed between measurements, indicating that the results were evaluated using a mixed quantitative and qualitative method not mentioned in the methodology. The nature of this method should be specified.
Response 4: This is already stated in lines 158 to 159.
Comments 5: Was any statistical test used to assess the quantitative differences between the two Family Spirituality Index measurements? How were qualitative changes assessed?
Response 5: Since these are scores for each family, no statistical analysis has been performed (it is not possible). The effect of the FLR was assessed using a mixed method of changes in scores and the results of semi-structured interviews.
Comments 6: The results are presented in detail and in tables that facilitate understanding, although it should be clarified why not all items of the Family Spirituality Index appear in the tables. Why were the ones that did appear chosen? Why do items 17 through 21 not appear in any of the tables?
Response 6: This is already stated in lines 226 to 227.
Comments 7: The discussion qualitatively analyzes the results obtained and establishes relationships with previous studies; although at no point does it perform a quantitative analysis specifying which results obtained after the intervention was statistically significant. If the results were assessed quantitatively, as stated in the tables, the significance of the changes produced by the intervention should be indicated.
Response 7: Since these are scores for each family, no statistical analysis has been performed (it is not possible). The effect of the FLR was assessed using a mixed method of changes in scores and the results of semi-structured interviews.
Comments 8: I believe that the limitations should indicate those arising from the type of sample used.
Response 8: In lines 506 to 508, it has been stated that there are limitations to the generalizability of the study results because the study targeted families including older adults. To facilitate comprehension, we have modified the passage below.
This study recruited families at two visiting nurse stations in a city in Japan, and targeted a small number of families that included older adults, thereby limiting the generalizability of the study results.
Comments 9: I imagine the conclusions are consistent with the results obtained, although to do so I would need to know the method used to assess the results and the statistical significance of the quantitative changes that occurred in the Family Spirituality Index after the intervention.
The references are adequate and mostly up-to-date, allowing the reader to delve deeper into the research question.
Response 9: The following has been added in lines 515 to 517.
The effects of FLR were evaluated using a mixed-methods approach using a semi-structured interview regarding family spirituality and scores from the Family Spirituality Index.
Round 2
Reviewer 1 Report
Comments and Suggestions for Authors
As reviewer I have no conflicts of interest with this work.
Detailed Article Report
Article Title: The Effects of Family Intervention Utilizing Family Life Review on Changes in Family Spiritual Suffering
Authors: Naohiro Hohashi and Haruka Yano
Brief summary
The study investigates the impact of a family intervention using family life review (FLR) on changes in family spirituality in families experiencing spiritual suffering, through a quasi-experimental study. The results show that FLR can maintain, enhance, confer and update family spirituality, becoming a legacy for the family and increasing awareness of the value of its existence. As strengths, the paper addressed an underexplored topic, it is based on an adequate nursing theoretical model or uses triangulation and discussion between evaluators to reduce the risk of bias in interviews.
General concept comments
The present version includes the recommendations made in the first review, improving the quality of the work presented. I want to congratulate the authors for their effort.
As a final comment on the changes made, I just wanted to offer a small suggestion regarding the expression "improvement", which has replaced "effectiveness". While “improvement” is grammatically correct, it might not fully capture the intended meaning. “…to verify its improvement as family spirituality support.” (line 95) and “…demonstrating its improvement as a method…” (line 520) suggests that the FLR has improved over time. But if what you mean is that FLR proves to be valuable, then "improvement" could not be the best option.
If the goal is to highlight FLR’s value as a method, then a more precise term could be value, utility or usefulness.
Author Response
Thank you for your comment.
On lines 95 and 519, we have changed improvement to usefulness.
Reviewer 2 Report
Comments and Suggestions for Authors
Dear Authors,
I consider that the manuscript has been sufficiently improved and I have no additional comments or suggestions.
Kind regards.
Author Response
To the reviewer: Thank you for your comments, which are much appreciated.